# OPTIMIZING MPJPE PROMOTES MISCALIBRATION IN MULTI-HYPOTHESIS HUMAN POSE LIFTING

**Paweł A. Pierzchlewicz**[1,2], **Mohammad Bashiri**[1,2], **R. James Cotton**[3,4], **Fabian H. Sinz**[1,2,5,6]

[1]Institute for Bioinformatics and Medical Informatics, Tübingen University, Tübingen, Germany
[2]Department of Computer Science, Göttingen University, Göttingen, Germany
[3]Shirley Ryan AbilityLab, Chicago, IL, USA
[4]Department of Physical Medicine and Rehabilitation, Northwestern University, Evanston, IL, USA
[5]Department of Neuroscience, Baylor College of Medicine, Houston, TX, USA
[6]Center for Neuroscience and Artificial Intelligence, Baylor College of Medicine, Houston, TX, USA
{ppierzc,bashiri,sinz}@cs.uni-goettingen.de
rcotton@sralab.org

## ABSTRACT

Due to depth ambiguities and occlusions, lifting 2D poses to 3D is a highly ill-posed problem. Well-calibrated distributions of possible poses can make these ambiguities explicit and preserve the resulting uncertainty for downstream tasks, thus providing the necessary trustworthiness in safety-critical domains. This study shows that multi-hypothesis pose estimation methods produce *miscalibrated* distributions. We identify that miscalibration can be attributed to the optimization of mean per joint position error (MPJPE). In a series of simulations, we show that minimizing minMPJPE, the MPJPE of the best hypothesis, converges to the correct mean prediction. However, it fails to correctly capture the uncertainty, hence resulting in a miscalibrated distribution[1].

The task of estimating the 3D human pose (HPE) from 2D keypoints, known as lifting (Martinez et al., 2017), is an ill-posed problem. Therefore, some methods generate multiple hypotheses of 3D poses to account for these ambiguities (Li & Lee, 2019; Sharma et al., 2019; Biggs et al., 2020; Oikarinen et al., 2020; Li & Lee, 2020; Kolotouros et al., 2021; Wehrbein et al., 2021; Holmquist & Wandt, 2022; Pierzchlewicz et al., 2022). Many of these approaches implicitly estimate the conditional distribution $p(\mathbf{X} \mid \mathbf{C})$ of 3D poses $\mathbf{X}$ given the 2D keypoints $\mathbf{C}$ through sample-based methods. Since direct likelihood estimation in sample-based methods is usually not feasible, different sample-based evaluation metrics have become popular. We

Table 1: Comparison of different multi-hypothesis 3D human pose estimation methods on the Human3.6M dataset and the corresponding expected calibration error (ECE).

| | Method | H36M (mm) | ECE |
|---|---|---|---|
| | Martinez et al. (2017) | 62.9 | - |
| | Zhao et al. (2019) | 60.8 | - |
| MPJPE | Gaussian (minMPJPE) | 54.8 | 0.42 |
| | Gaussian (NLL) | 60.1 | 0.07 |
| | Sharma et al. (2019) | 46.7 | 0.36 |
| | Oikarinen et al. (2020) | 46.2 | 0.16 |
| | Wehrbein et al. (2021) | 44.3 | 0.18 |
| NLL | Kolotouros et al. (2021) (GT) | 37.1 | 0.07 |
| | Pierzchlewicz et al. (2022) | 53.0 | 0.08 |

show that minMPJPE, the most commonly used metric, encourages overconfident distributions rather than correct estimates of the true distribution. As a result, it does not guarantee that the estimated density of 3D poses is a faithful representation of the underlying data distribution and its ambiguities.

## 1 MPJPE PROMOTES MISCALIBRATION

The inherent ambiguities of the lifting problem result in a posterior distribution $p(\mathbf{X} \mid \mathbf{C})$ that captures the unavoidable uncertainty. For the approximate posterior $q(\mathbf{X} \mid \mathbf{C})$ to be calibrated means it should assign the same probability to an event as the true posterior (Brier, 1950). Naeini et al. (2015) propose to measure the expected calibration error (ECE) which approximates the ex-

---

[1]Code is available at https://github.com/sinzlab/cgnf.

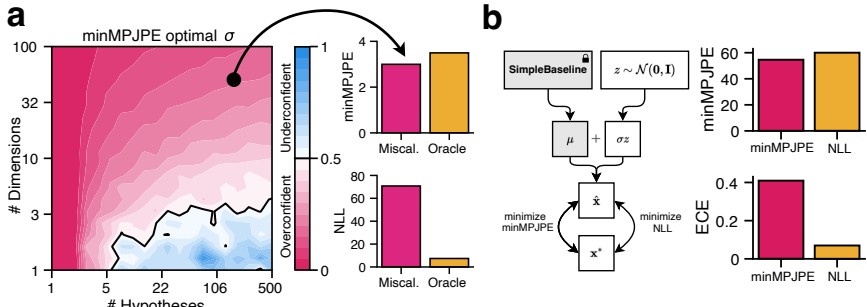

Figure 1: **a**) Standard deviation $\boldsymbol{\sigma}$ of a Gaussian distribution that minimizes minMPJPE for different numbers of samples and dimensions. The true $\boldsymbol{\sigma}^* = 0.5$ (black line), **underconfident** $\boldsymbol{\sigma} > \boldsymbol{\sigma}^*$ (blue), **overconfident** $\boldsymbol{\sigma} < \boldsymbol{\sigma}^*$ (pink). The human pose equivalent distribution (black point, 45 dimensions, 200 samples) compared to an oracle distribution (with true $\mu$ and $\sigma$) in terms of minMPJPE and NLL. **b**) (left) Gaussian noise baseline schematic. Optimizes two objectives 1) minMPJPE and 2) NLL between the predicted pose distribution $\hat{\mathbf{x}}$ and the ground truth 3D pose $\mathbf{x}^*$. The SimpleBaseline model weights are frozen. Right, comparison of the performance on minMPJPE and ECE when optimizing for minMPJPE and NLL.

pected value of the absolute difference between the predicted probability and the true probability: $\text{ECE} = \frac{1}{N}\sum_{n=1}^{N}|q_n - p_n|$, where lower ECE indicates a better-calibrated distribution. Here, we provide an algorithm for computing ECE in the pose estimation domain (See Section A.3).

We evaluate the approximate posterior achieved by minimizing minMPJPE for samples from a known true posterior. We consider a true posterior $p = \mathcal{N}(\boldsymbol{\mu}^*, \boldsymbol{\sigma}^{*2}\boldsymbol{I})$ with known mean and variance and an approximate posterior $q = \mathcal{N}(\boldsymbol{\mu}^*, \boldsymbol{\sigma}^2\boldsymbol{I})$. We optimize $\boldsymbol{\sigma}^2$ for different numbers of dimensions and hypotheses such that minMPJPE is minimized. Minimizing minMPJPE leads to a miscalibrated distributions (Fig. 1a). In the HPE domain (black point in Fig. 1a) the minMPJPE-optimal distribution (*Miscal.*) is miscalibrated and has a high negative log-likelihood (NLL), as opposed to the NLL-optimal distribution (*Oracle*) For analitical evidence see sections A.1 and A.2.

## 2 GAUSSIAN NOISE BASELINE ON HUMAN 3.6M

To evaluate the conclusions of section 1 in a real-world scenario, we use a Gaussian model that can be optimized for both minMPJPE and NLL on the Human3.6M dataset (Catalin Ionescu, 2011; Ionescu et al., 2014) (for data details see sec. B). We train an additive Gaussian noise model on top of the pre-trained SimpleBaseline (Martinez et al., 2017) We generate $N$ pose hypotheses $\hat{\mathbf{X}}$ for $M$ observed 2D poses $\mathbf{C}$ according to $\hat{\mathbf{X}}_{n,m} = \text{SimpleBaseline}(\mathbf{C}_m) + \boldsymbol{\sigma}\boldsymbol{z}_n$ where *only* the variance $\boldsymbol{\sigma}$ is a trainable parameter, while the mean is fixed and $\boldsymbol{z} \sim \mathcal{N}(\boldsymbol{z}; \mathbf{0}, \boldsymbol{I})$ (Fig. 1b). We optimize for either minMPJPE or NLL. We observe the minMPJPE model to be overconfident with lower minMPJPE than the better-calibrated NLL model. Conversely, equally accurate but calibrated methods can seem inferior due to minMPJPE measuring *not only* accuracy, but also precision.

We furthermore, test the calibration of a few state-of-the-art multi-hypothesis methods and find that all the methods that optimize towards a sample-based objective are miscalibrated while, optimizing for NLL provides a well-calibrated distribution (Tab. 1).

## 3 CONCLUSION

In this study, we provide evidence that the focus on achieving the lowest minMPJPE in multi-hypothesis 3D pose estimation is leading to miscalibrated distributions. We identify that optimizing for distribution-based objectives like NLL leads to well-calibrated distributions. We believe that our findings will be useful for identifying and mitigating miscalibration in multi-hypothesis pose estimation and will lead to more robust and safer applications of multi-hypothesis pose estimation.

ACKNOWLEDGMENTS

We thank Alexander Ecker, Pavithra Elumalai, Arne Nix, Suhas Shrinivasan and Konstantin Willeke for their helpful feedback and discussions.

Funded by the German Federal Ministry for Economic Affairs and Climate Action (FKZ ZF4076506AW9). This work was supported by the Carl-Zeiss-Stiftung (FS). The authors thank the International Max Planck Research School for Intelligent Systems (IMPRS-IS) for supporting MB.

URM STATEMENT

Author PP meets the URM criteria of the ICLR 2023 Tiny Papers Track.

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

## A METRICS

### A.1 MEAN PER JOINT POSITION ERROR

A popular optimization metric is the MPJPE. While this metric is especially popular in single-pose estimation methods, it has also been used in various forms in multi-hypothesis methods. Optimizing this metric causes the distribution of poses to be overconfident. We show this for a simple one-dimensional distribution, the generalization to the multi-dimensional case is straightforward. Given samples $x \sim p(x|c)$ from a data distribution given a particular context $c$, such as keypoints from a image, consider an approximate distribution $q(\hat{x}|c)$ supposed to reflect the uncertainty about $x|c$.

This below objective is equivalent to the mean position error for a single joint. Note that $x$ and $\hat{x}$ are conditionally independent given $c$, i.e. $x \perp \hat{x}|c$. The objective can then be expanded as follows:

$$
\begin{aligned}
\mathcal{L} &= \mathbb{E}_{x \sim p(x|c), \hat{x} \sim q(\hat{x}|c), c} \left[ (x - \hat{x})^2 \right] \\
&= \mathbb{E}_c \left[ \mathbb{E}_{x, \hat{x}|c} \left[ (x - \mu_c + \mu_c - \hat{x})^2 \right] \right] \\
&= \mathbb{E}_c \left[ \underbrace{\text{Var}[x \mid c]}_{\text{indep. of } q} - 2\mathbb{E}_{x, \hat{x}|c} \left[ (x - \mu_c)(\hat{x} - \mu_c) \right] + \mathbb{E}_{\hat{x}|c} \left[ (\hat{x} - \mu_c)^2 \right] \right] \\
&= \text{const.} - 2\mathbb{E}_c \left[ \underbrace{\mathbb{E}_{x|c} \left[ (x - \mu_c) \right]}_{=0} \mathbb{E}_{\hat{x}|c} \left[ (\hat{x} - \mu_c) \right] + \mathbb{E}_{\hat{x}|c} \left[ (\hat{x} - \mu_c)^2 \right] \right] \\
&= \text{const.} + \mathbb{E}_c \left[ \mathbb{E}_{\hat{x}|c} \left[ (\hat{x} - \mu_c)^2 \right] \right] \geq 0
\end{aligned}
$$

The expectation in the final line is non-negative and can be minimized by $q(\hat{x}|c) = \delta(\hat{x} - \mu_c)$, i.e. setting $\hat{x} = \mu_c$ and shrinking the variance to zero. This means that $q$ would be extremely overconfident.

### A.2 minMPJPE CONVERGES TO THE CORRECT MEAN

Consider 1D samples $x^*$ from a data distribution $p(x)$ and an approximate Gaussian distribution $q(x)$ with parameters $\mu$ and $\sigma$. We sample $N$ hypotheses from $q(x)$ and minimize the minMPJPE objective:

$$
\text{minMPJPE} = \mathbb{E}_{q(z)} \left[ \mathbb{E}_{p(x)} \left[ \min_i (x^* - \mu - \sigma z_i)^2 \right] \right]
$$

Consider $z_j^*$ as the $z_i$ sample which minimizes the expression for the $j$-th data sample $x_j^*$.

$$
\text{minMPJPE} = \mathbb{E}_{q(z)} \left[ \mathbb{E}_{p(x)} \left[ (x^* - \mu - \sigma z_j^*)^2 \right] \right]
$$

Thus the derivative can be computed to be

$$
\frac{\partial}{\partial \mu} \text{minMPJPE} = -2\mathbb{E}_{q(z)} \left[ \mathbb{E}_{p(x)} \left[ x^* - \mu - \sigma z_j^* \right] \right] = 0
$$

$$
= \mathbb{E}_{p(x)} \left[ x^* \right] - \mu - \mathbb{E}_{q(z)} \left[ z_j^* \right]
$$

Simulations indicate that $\mathbb{E}_{q(z)} \left[ z_j^* \right]$ can be approximated by a sigmoid function $S : \mathbb{R} \mapsto [-1, 1]$ with $S(0) = 0$.

$$
\mathbb{E}_{q(z)} \left[ z_j^* \right] = S \left( \mathbb{E}_{p(x)} \left[ x^* \right] - \mu \right) \cdot C(\sigma, N)
$$

where $C(\sigma, N)$ is a scalar scaling value dependent on $\sigma$ and the number of hypotheses. Thus the root of the derivative can be computed to be:

$$
\mu = \mathbb{E}_{p(x)} \left[ x^* \right]
$$

## A.3 QUANTILE CALIBRATION FOR POSE ESTIMATION

---

**Algorithm 1** Quantile calibration for pose estimation

---

**for** each $\mathbf{X}_m^*$ and $\mathbf{C}_m$ **do**
    draw $N$ hypotheses $\hat{\mathbf{X}} \mid \mathbf{C}_m$
    $\tilde{\mathbf{X}}_{m,k} \leftarrow \text{CTM}(\hat{\mathbf{X}}_{:,m,k})$                  $\triangleright$ Central Tendency Measure
    $\varepsilon_{n,m,k} \leftarrow ||\hat{\mathbf{X}}_{n,m,k} - \tilde{\mathbf{X}}_{m,k}||_2$
    $\Phi_m(\varepsilon) \leftarrow \text{CDF}(\varepsilon_{:,m,k})$
    $\varepsilon_{m,k}^* \leftarrow ||\mathbf{X}_{m,k}^* - \tilde{\mathbf{X}}_{m,k}||_2$
**end for**
$\omega_k(q) \leftarrow \frac{1}{M} \sum_{m=1}^M \mathbf{1}_{\Phi_m(\varepsilon_{m,k}^*) \leq q}$
$\omega(q) \leftarrow \text{median}(\omega_k(q))$
$\text{ECE} = \frac{1}{|\mathcal{Q}|} \sum_{q \in \mathcal{Q}} |\omega(q) - q|$

---

Quantile calibration (Song et al., 2019) defines a perfectly calibrated distribution as one for which ground-truth values $\mathbf{X}^*$ fall within the $q$-th quantile $q\%$ of the time. However, for high dimensions estimating whether a point is contained within a given quantile is non-trivial. We, therefore, propose to simplify the problem by projecting to the univariate space of squared errors $\varepsilon$ from the central tendency $\tilde{\mathbf{X}}$ of $N$ hypotheses $\hat{\mathbf{X}}$ conditioned on 2D poses $\mathbf{C}$ with $K$ keypoints. We then compute ECE in the space of $\varepsilon$ over the set of quantiles $\mathcal{Q} \in [0,1]$ (Algorithm 1). As a measurement of central tendency we choose the median statistic, which is more robust to outlier samples. However, in practice, the choice of median vs. mean results in minor differences in the calibration outcomes as we show in sec. A.4.

## A.4 IMPACT OF THE CENTER TENDENCY MEASURE ON EXPECTED CALIBRATION ERROR

The choice of center tendency measure should be considered when computing the expected calibration error. Therefore on a subset of the models presented in table 1 we compare the effect of choosing 3 different reference points. 1) The median of the samples 2) the mean of the samples and 3) the mode of the samples. We showcase the results in table 2. We observe that the use of median

Supplementary Table 2: Comparison of different reference points definitions on the resulting ECE score. In bold we mark the method that under the particular reference point has the lowest ECE.

| Method | Median | Mean | Mode |
|---|---|---|---|
| Sharma et al. (2019) | 0.36 | 0.36 | 0.14 |
| Wehrbein et al. (2021) | 0.18 | 0.18 | 0.08 |

in contrast to mean has little to no effect on the computation of ECE. Using the mode as a reference point results in generally smaller values of ECE.

## B DATA

We use the Human3.6M Dataset (H36M) on the academic use only license (Catalin Ionescu, 2011; Ionescu et al., 2014) which is the largest dataset for 3D human pose estimation. It consists of tuples of 2D images, 2D poses, and 3D poses for 7 professional actors performing 15 different activities captured with 4 cameras. Accurate 3D positions are obtained from 10 motion capture cameras and markers placed on the subjects. We evaluate the models on every 64th frame of subjects 9 and 11.

