# OpenReview forum: "Optimizing MPJPE promotes miscalibration in multi-hypothesis human pose lifting"
_ICLR.cc/2023/TinyPapers — Submitted to Tiny Papers @ ICLR 2023_

### Official Review · Reviewer_1awW · 2023-03-23

**Confidence:** 2

**Summary Of Contributions:**

The authors aim to demonstrate that the standard approach for pose lifting (from 2D to 3D), by minimization of the MPJPE loss, leads to overconfident models, that do not represent correctly the uncertainty of this ill-posed problem. Through experiments and formal demonstrations, the paper shows that minMPJPE generates lower-variance models (overconfident), when compared to a model trained with Negative Log Likelyhood, even though the mean error is lower (better).

**Rating:**

Clear, Correct, and Reproducible (CCR): a submission which meets the reviewing criteria

**Strengths And Weaknesses:**

### Strengths
1. The paper is clearly written and easy to follow.
1. The experiments support the hypothesis presented.
1. The figure is clear and well designed.
1. The work seems to be reproducible using the information in the main paper.
### Weaknesses
1. In fig.1, $x^*$ is not explained. By context I assume it's the Ground Truth 3D pose, but a clarification is needed.

**Suggested Changes:**

1. See weaknesses

---

> ### Author Response · Authors · 2023-05-31
> **Response to Reviewer 1awW**
>
> We appreciate that you find our paper to be well-written, easy to follow, and our experiments support our hypothesis. To further aid with reproducibility, we have made the code for reproducing the results publicly available.
>
> Regarding the missing explanation of $x^*$ in the caption of Fig. 1. You are correct in inferring that it refers to the ground truth pose. We have added that explanation to the caption.

---

### Official Review · Reviewer_345q · 2023-03-29

**Confidence:** 4

**Summary Of Contributions:**

In this paper, the authors study the problem of miscalibration in human pose lifting when utilising the Mean Per-Joint Prediction Error as a loss function. The authors provide both simluations using a simple baseline and experiments using well-known human pose estimation models, showing the issue of miscalibration in practice.

**Rating:**

High Potential (HP): a submission which meets the reviewing criteria and has potential to make an impact on the field

**Strengths And Weaknesses:**

**Strengths**

- The paper is well-written and easy to follow
- The authors have provided both a baseline and results on well-known models in HPE to validate their hypothesis
- The paper demonstrates that it is worth looking at lifting models from a probabilistic perspective, especially when considering multiple-hypothesis models. This opens up new possibilities for the study of HPE, which often is thought of as a purely geometrical problem.
- The proposed finding easily opens up models that maximize the likelihood of the data explicitly, such as VAEs, Flows, or Diffusion models. I believe that the findings in this paper can be a great starting point for people looking to implement probabilistic solutions for HPE

**Weaknesses**

- The claim of miscalibration is definitely important, but in the end, HPE is a geometric problem whose solution is measured as the distance of predicted and ground-truth joints in 3D space. That being the case, MPJPE provides a good measure of the "accuracy" of the estimation. Perhaps the real problem is the fact that this measure is the target to be optimized and that numerous papers will directly optimize it to maximize performance on different benchmarks. Nevertheless, having a correct mean prediction is a very important aspect, something which as seen in Table 1 (Gaussian MPJPE vs NLL), simple NLL-based models fail to do as well. I believe the authors could have provided a couple of sentences regarding this aspect

**Suggested Changes:**

I believe that if the authors could address the only weakness listed in the review, which already contains some suggestions, the paper would definitely be more complete.

---

> ### Author Response · Authors · 2023-05-31
> **Response to Reviewer 345q**
>
> We appreciate that you find that our paper is well-written and easy to follow. We are glad that you agree that the probabilistic perspective on human pose estimation is worth looking at.
>
> We agree that HPE is inherently a geometric problem. Section 2 shows, however, that minMPJPE is not a measure of accuracy. The model in Section 2 uses the same fixed mean (output of SimpleBaseline) and only the noise variance is optimized. Nevertheless, minMPJPE optimizes variance to an overconfident value. As you have accurately observed the problem in the field lies in using minMPJPE as the key metric. As per our findings, is not only a measure of accuracy but is also affected by precision. As a result, calibrated methods that are equally accurate can seem inferior in terms of minMPJPE.
>
> To address your comment, we made it clearer in section 2 that the mean is kept constant between the two optimization paradigms and added a sentence relating the results to the quality of minMPJPE as a metric of accuracy.

---

### Comment · Area_Chair_suJg · 2023-05-30
**Inviting to archive**

To the authors,

Congratulations on having your paper chosen as ``notable''.

I notice you haven't de-anonymized the PDF or opted-in for archiving.  I encourage the authors to at least de-anonymize the paper -- to make sure your names as authors are always on any public version of the work --  as the OpenReviews + PDFs will be made public.

Archiving the paper will result in a DOI being allocated, which you may reasonably not want if you are looking to subsequently publish.  You can de-anonymize the PDF without archival.

Thanks,
AC suJg

---

> ### Author Response · Authors · 2023-05-31
> **Archival Opt-in**
>
> Thank you.
>
> We have de-anonymized the paper and responded to the reviews.
>
> We wish to opt-in for archival.

---

> > ### Comment · Area_Chair_suJg · 2023-06-02
> > **Camera-ready Formatting**
> >
> > Thank you for opting in for publication, de-anonymising, and including the URM statement.  The manuscript is currently over the two-page length limit.  Please can the authors re-upload a version that is under two pages.  The URM and acknowledgements do not need to fit inside the two pages.
> >
> > Thank you very much!
> >
> > AC suJg

---

> > > ### Author Response · Authors · 2023-06-03
> > > **Camera-ready Formatting**
> > >
> > > Re-uploaded a version that is under 2 pages.
> > > Sorry, we thought that the 2-page limit did not include the author list.

---

### Comment · Area_Chair_suJg · 2023-06-06
**Ready for archival**

This work meets the threshold for archival, contents the URM statement and is deanonymized.

---

### Meta-Review · Area_Chair_suJg · 2023-04-02

**Recommendation:** Invite to present (notable)
**Confidence:** 4

**Metareview:**

This paper studies the loss metric optimised in 2d->3d video reconstruction as a potential source of mis-calibration.  This is an important topic, as this misquantmfication of uncertainty may be catastrophic for downstream tasks. The work is clearly written, appears to be novel, and dovetails nicely with recent work.  The paper itself is very nicely prepared.  Although the example and "method" are quite simple, I think it elegantly highlights a potential problem and presents a very tractable solution that will engender follow-up work.  As such, I believe this work is significant and is worthy of inclusion.

**Summary:**

Existing loss metrics and training methods yield overconfident models.  Instead, optimising the NLL better calibrates the uncertainty.

**Comments And Feedback To The Authors:**

I hope the authors implement the minor changes requested by the two reviewers.

My only question is whether this correction can be applied to any prediction method?  For instance, could the Sharma estimator be "corrected" using this method.

I also look forward to the authors releasing code for this work.

**Reason For Not Giving A Higher Recommendation:**

N/A

**Reason For Not Giving A Lower Recommendation:**

I believe the paper very neatly motivates, analyses, and shows a (albeit simple) example solution to a problem, alongside sensible other comparisons.  I believe one of the reviewers ratings was a touch low (maybe why the reviewer also listed a low confidence), and therefore give preference to the HP score given by 345q, and my own evaluation (HP).

---

> ### Author Response · Authors · 2023-05-31
> **Response to Area Chair suJg**
>
> We appreciate that you consider the topic of our study to be important and that you find that our experiments elegantly highlight the potential problem.
>
> We have addressed the comments of the reviewers **345q** and **1awW** in a revised version of the manuscript and comments.
>
> Regarding your question. In principle, not all predictors can be optimized using NLL, especially, for example, in the case of Sharma et al. who use additional metrics, which do not easily translate to NLL. Although our findings might not allow for the correction of any arbitrary method, it allows identifying which methods should result in miscalibrated distributions.
>
> The code is available now at https://github.com/sinzlab/cgnf

---

### Decision · Program_Chairs · 2023-04-10

Invite to present (notable)